# Combined Ultrahypofractionated Whole-Breast Irradiation and IORT-Boost: A Safety and Feasibility Analysis

**DOI:** 10.3390/cancers16061105

**Published:** 2024-03-09

**Authors:** Javier Burgos-Burgos, Víctor Vega, David Macias-Verde, Virginia Gómez, Elena Vicente, Carmen Murias, Carlos Santana, Pedro C. Lara

**Affiliations:** 1Canarian Comprehensive Cancer Center, San Roque University Hospital, 35001 Las Palmas de Gran Canaria, Spain; javier.burgos@hospitalessanroque.com (J.B.-B.); victor.vega@hospitalessanroque.com (V.V.); david.macias@hospitalessanroque.com (D.M.-V.); virginia.gomez@hospitalessanroque.com (V.G.); elena.vicente@hospitalessanroque.com (E.V.); camen.murias@hospitalessanroque.com (C.M.); carlos.santana@hospitalessanroque.com (C.S.); 2Department of Medicine, Fernando Pessoa Canarias University, 35001 Las Palmas de Gran Canaria, Spain; 3Canarian Institute for Cancer Research, 38204 San Cristobal de La Laguna, Spain

**Keywords:** ultrahypofractionation, breast cancer, photon-IORT, boost

## Abstract

**Simple Summary:**

The aim of the present study is to assess, for the first time, the feasibility and safety of combining photon IORT with ultrahypofractionated WBI after breast-conserving surgery (BCS). From July 2020 to December 2022, seventy-two patients diagnosed with low-risk early BC were included in this prospective study. IORT was administered at a dose of 20 Gy to the surface’s applicator, and WBI was to be administered 3–5 weeks after surgery at a total dose of 26 Gy in five consecutive days. All patients completed the proposed treatment schedule, and no severe acute or late grade 3 toxicity was observed at 3 and 12 months after WBI, respectively. Our results confirm, for the first time, that the combination of ultrafractionation WBI after BCS and photon-IORT is a feasible and safe option in patients with early BC.

**Abstract:**

Background: The current standard of local treatment for patients with localized breast cancer (BC) includes whole breast irradiation (WBI) after breast-conserving surgery (BCS). Ultrahypofractionated WBI schemes (1-week treatment) were shown not to be inferior to the standard WBI. Tumor bed boost using photon intraoperative radiotherapy (IORT) is safe and feasible in combination with standard WBI. The aim of the present study is to assess, for the first time, the feasibility and safety of combining photon IORT with ultrahypofractionated WBI. Methods: Patients diagnosed with low-risk early BC candidates for BCS were included in this prospective study. IORT was administered at a dose of 20 Gy to the surface’s applicator, and WBI was administered 3–5 weeks after surgery at a total dose of 26 Gy in five consecutive days. Results: From July 2020 to December 2022, seventy-two patients diagnosed with low-risk early BC and treated in our institution were included in this prospective study. All patients completed the proposed treatment, and no severe acute or late grade 3 toxicity was observed 3 and 12 months after WBI, respectively. Conclusions: Our results confirm for the first time that the combination of ultrafractionation WBI and photon-IORT after BCS is a feasible and safe option in patients with early BC.

## 1. Introduction

The current standard of local treatment for patients with localized breast cancer (BC) includes breast conservative surgery (BCS) associated with whole breast irradiation (WBI) [1]. This treatment approach has been shown to be equivalent to mastectomy in terms of local control and survival [2,3,4]. Although conventionally fractionated radiotherapy (50 Gy 2 Gy per fraction) administered in 25 fractions in 5 weeks was historically used, clinical evidence from moderate hypofractionation trials [5,6,7] with shorter treatment times and higher doses per session, based on the low alpha/beta (3 Gy) of BC [8], gave favorable results. These moderately hypofractionated schemes included 40.05 Gy at 2.67 Gy per fraction during a total of 15 fractions administered over 3 weeks. The increased local control and reduced toxicity rates observed in moderate hypofractionation trials [5,6,7] encouraged researchers to try to further increase the doses per session, further shortening the total treatment time. In May 2020, the results of the ultrafractionation Fast Forward trial [9] were released, showing that the 26 Gy/5.2 Gy per fraction, during five fractions administered in only 1 week, scheme was not inferior to the standard 40.05 Gy/15 fx/3 w for local tumor control and is equally safe in terms of effects on normal tissue.

Unfortunately, most local recurrences of BC have been observed to occur at the original primary site [10,11,12]. An additional dose of radiation to the tumor bed (boost radiation dose) is accompanied by an increase in local control for patients, at the expense of greater toxicity, depending on the technique used [13]. According to this evidence, WBI + boost dose would be considered the standard treatment for breast cancer patients after BCS [13].

In the FAST Forward trial, 25% of patients received a sequential boost of five to eight fractions of 2Gy. This treatment was well tolerated, but it increased the total treatment time [9]. In these cases, photon-intraoperative radiotherapy (IORT) as a technique to administer this “boost” would be an attractive approach, allowing the administration of a single high dose of radiation in the same surgical procedure with direct visualization of the surgical bed (avoiding its geographic loss prior to oncoplastic manipulation), saving total treatment time and achieving significant savings in doses to the skin [14]. The safety and feasibility of the combination of moderate hypofractionation WBI and IORT boost have already been assessed by our group [15,16].

The aim of this study is to assess, for the first time, the safety and feasibility of a combined ultrafractionation WBI and IORT boost in patients treated with BCS in our institution.

## 2. Materials and Methods 

Patients diagnosed with low-risk early BC and candidates for BCS were included in this prospective study. Patients were diagnosed and treated at the Canarian Comprehensive Cancer Center, San Roque University Hospital in Las Palmas de Gran Canaria, Spain. In all cases, the approval of the Ethics Committee and written informed consent were obtained. Patient´s inclusion criteria were: age 18 years or older, histologically and radiologically proven early BC, tumor size less than 2 cm, Luminal molecular profile, and must be discussed in Multidisciplinary Tumor Boards (MTB) of our institution. Exclusion criteria were: age less than 18 years, histological types other than epithelial breast cancer, axillary positive nodes, locally advanced or metastatic disease, or ECOG (Eastern Cooperative Oncology Group) > 3. Cancer staging was performed according to the 8th edition of the TNM classification system [17].

WBI was prescribed to patients ≤ 45 years of age and those who presented in their pathology report infiltrative lobular histology, pure intraductal carcinoma in situ, extensive intraductal component, lymphovascular invasion and close or affected surgical margins.

From July 2020 to December 2022, seventy-two patients were included in the study. The clinical characteristics of the patients are shown in Table 1. The median age was 56 years (range 36–72 years). The size of the tumor was less than 2 cm in all cases. Thirty-two patients presented with premenopausal status, and forty patients with postmenopausal status received the corresponding endocrine therapy according to the international guidelines [18,19]. No patient was considered for chemotherapy.

### 2.1. Treatment Description

All patients were referred to BCS and immediate IORT at the time of lumpectomy. Hormonal therapy was prescribed according to international guidelines [18,19]. The IORT treatment in our institution has been previously described [15,16]. In short, a radiation dose of 20 Gy was prescribed to the surface of the applicator [15,16] of a portable low-energy X-ray accelerator (50 kv), Intrabeam^®^ (Carl-Zeiss Ober Kochen Germany). Postoperative whole breast irradiation (WBI) was administered to patients showing the TARGIT A adverse factors present in the pathology report [20]. The WBI dose to be administered was that of the ultrafractionation scheme (26 Gy/5.2 Gy/5 fx) of the Fast Forward study [9], and it was planned to be administered 3 to 5 weeks after BCS.

WBI treatment after IORT in our institution has been previously described [15,16]. All patients were treated by the VMAT-Rapid Arc technique. CTVs and OARs contouring were based on the recommendations of RTOG [21]. The same OARs constraints as the reference FAST-FORWARD study were used [9]. The plans were optimized for Rapid Arc delivery, and the PRO algorithm [22] was used to modulate the shape of the multi-leaf collimator and the intensity of the beam during gantry rotation. Dose calculations used the anisotropic analytical algorithm (AAA).

A VARIAN Linac made the delivery using 6 MV photons (Palo Alta, CA, USA). Daily image-guided radiation therapy (IGRT) using an Exac-Trac System (Brain Lab, Munich, Germany) was performed on all patients.

In addition, all patients were offered to receive a comprehensive program of nutrition counseling, physical rehabilitation, psycho-oncology support and dermo-aesthetic therapy with the aim of improving functionality and the patient´s quality of life.

### 2.2. Evaluation

The defined primary end points of the study were the safety and feasibility of the proposed treatment protocol. Safety was evaluated by the rate of grade 3 acute and late toxicity observed in the treated patients, according to the common toxicity criteria for adverse effects (CTCAE 5.0) [23]. Patients were evaluated for toxicity one, three and six months after the end of radiotherapy. Patients were followed prospectively under the Spanish RD1566/1998 regulation for Radiation Therapy Quality Assurance.

Follow-up visits were subsequently scheduled every 3 months during the first two years and every six months, thereafter. Breast ultrasound and mammograms were performed six months after BCS and yearly thereafter, unless otherwise recommended. Follow-up was jointly performed by the staff doctors of the departments of surgery and oncology at the Canarian Comprehensive Cancer Center.

The treatment feasibility was evaluated through the number of patients that completed the scheduled treatment and the number of patients that suffered interruptions of the treatment.

The defined secondary endpoints included cosmetic results evaluated by the Harvard scale [24], local relapse rate, and cause-specific/overall survival.

## 3. Results

### 3.1. Feasibility

All 72 patients received the scheduled boost of 20 Gy with photon-IORT at the time of the lumpectomy. The average diameter of the spherical applicator was 3 cm (2.5–3.5 cm). All patients received the ultrafractionation WBI scheme. Sixty-three out of the 72 included in the study started WBI within the predefined time period (3–5 weeks after BCS). Patients who started WBI after the fifth week were those who (a) had surgical wound complications (5 patients, 6.9%) or (b) seromas that required intervention/drainage prior to starting radiotherapy (2 patients, 2.8%) and (c) those that required margin widening surgery (extensive margin infiltration, 2 patients, 2.8%). Fifty-nine patients (82%) started their treatment on Monday and finished it on Friday of the same week. For thirteen patients, it was not possible to administer the treatment Monday–Friday due to machine-slot scheduling availability. In these cases the total treatment time was 7 days.

### 3.2. Safety 

Acute toxicity was assessed between 1 and 3 months according to the CTCAE v5.0 scale [23] and is summarized in Table 2. Four patients showed grade 3 dermatitis and pruritus and two referred pain, one month after the end of WBI. No severe acute toxicity remains 3 months after the end of WBI.

Late toxicity was evaluated 6 months after completing radiotherapy and every six months thereafter (Table 3). No severe late toxicity (grade 3) was observed. During follow-up, one rib fracture was observed within the treatment field. G1-2 toxicity, such as fibrosis and skin edema, were the most frequently recorded.

Imaging findings in scheduled ultrasound/mammogram tests for asymptomatic patients occasionally showed residual seromas (11 patients, 15.2%) and fat necrosis (7 patients, 9.7%), not requiring medical intervention in any case.

Cosmetic results were evaluated from 6 months after completing radiotherapy onwards, following the Harvard scale [24]. Excellent/good outcome was classified in 76.3% of patients (55 patients), fair outcome in 16.7% (12 patients) and poor outcome in 7% (5 patients). It should be noted that asymptomatic residual seromas and fat necrosis detected in the follow-up imaging tests did not affect the breast architecture, which is why they were not considered for the cosmetic result by both medical and patient´s criteria.

Follow-up was closed on 1 December 2023. The median follow-up was 22 months (range 6–36 months). At the final follow-up visit, all patients were alive and without evidence of local, regional or distant disease.

## 4. Discussion

The objective of this study was to evaluate the combination of a high-dose “boost” with Intrabeam^®^ at the time of BCS and ultrafractionation WBI delivered with the VMAT-Rapid Arc technique in patients with early BC and determine its feasibility in relation to toxicity and aesthetics.

Shortening the total treatment time through moderate hypofractionation has been one of the major advances in BC treatment [5,6,7], while also improving patients’ frequentation at the treating center and their quality of life [25]. Unfortunately, when a tumor bed boost was needed, a conventionally fractionated boost was used [13]. A simultaneous Integrated Boost (48 Gy/3.2 Gy per fraction/15 fractions) in this particular clinical situation has also become popular [26,27,28].

Conventionally fractionated WBI (50 Gy in 5 weeks) combined with IORT photon boost was shown to be a standard alternative to other boost techniques [29]. Moderate hypofractionation WBI (40.05 Gy in 3 weeks) combined with an IORT boost has been confirmed to be safe and feasible using both electrons [30,31,32,33] and photons [15,16,34,35]. Our pioneering experience has shown that the combination of immediate IORT plus moderate HWBI is accompanied by favorable results in relation to chronic toxicity and aesthetic results, both in patients with early and locally advanced disease [15,16].

Given that sensitivity to BC fractionation is comparable to that of dose-limiting normal tissues, it is reasonable to consider whether giving too few high-dose sessions would be useful in BC [36]. The Fast Forward trial was found not to be inferior in terms of local control and toxicity compared to moderate hypofractionation WBI [37]. Again, when a boost dose to the tumor bed is needed, 25% of Fast Forward patients had a sequential boost with conventionally fractionated external radiotherapy (10–16 G at 2 Gy/day), which increased the total treatment time [9].

No data are available about the combination of ultrahypofractionated WBI and IORT boosts. At present, different approaches to boost treatment when using ultrafractionation schedules have been proposed (Table 4). The HAI5 trial [38] treated 95 patients with alternate day ultrafractionation WBI at breast/chest wall doses of 28.5 Gy/5.7 Gy plus simultaneous integrated boost (SIB) in 66% of patients at 32.5/6.5 Gy or 34.5/6.9 Gy, depending on margins. In those patients with axillary involvement, 27 Gy/5.4 Gy were administered. The authors assessed acute toxicity with a median follow-up of <6 months. A total of 17.6% had acute G2-3 toxicity in the SIB group, compared to 0% when SIB was not administered.

From the same group, the YO-HAI5 (Young-Old Highly Accelerated Irradiation in five fractions) trial randomized BC patients after WBI-conservative surgery at 28.5 Gy/5 fx and a SIB of 6.2 Gy in 12 days or moderate WBI 40.05 Gy/15 fx with a SIB of 3.12 Gy/day. The authors observed a significantly higher incidence of acute breast edema, breast pain, asthenia, and skin toxicity in those treated with moderate HWBI compared to ultrafractionation [39].

In a Spanish experience, Montero et al. [40] in 383 patients with BC were included in a prospective registry, treated with ultrafractionation WBI 26 Gy/5.2 Gy/5 fx plus SIB of 29 Gy/5.8 Gy/5 fx in 272 patients and 30–31 Gy/6–6.2 Gy/5 fx in 111 patients, depending on margin status. The median follow-up was 18 months (range 7–31). Acute toxicity was acceptable, with 182 (48%) and 15 (4%) G1 and G2, respectively. Breast edema was observed in nine patients (2%) G1 and two patients (0.5%) G2. No other acute toxicities were observed. Late toxicity was mild, grade 1 breast edema was observed in 6 patients (2%), grade 1 hyperpigmentation in 20 patients (5%); and grade 1 and 2 breast induration below the reinforcement region in 10 (3%) and 2 patients (0.5%), respectively. Patients well tolerated this scheme [40].

On the other hand, the role of sequential boost was studied by Machiels et al. [41] in 102 patients using a Fast-Forward scheme plus an immediate sequential boost of a single fraction of 6 Gy in patients with risk factors for recurrence. The incidence of G1 and G2 acute skin toxicity was 74% and 2.7%, respectively.

In our opinion, IORT boost is an excellent treatment when using moderate hypofractionation radiotherapy and is therefore worthy of being combined with an ultrafractionation WBI protocol that produces similar toxicity figures to the moderate hypofractionation protocols [29,30,31,32,33,34,35].

A potential new field of research using this combination would be the possibility of administering the WBI after BCS and before the start of chemotherapy in cases of high-risk early BC [18,19]. In our opinion, completing the local treatment within 3–4 weeks after BCS would allow for improved the integration with systemic therapies, which would include new drugs for high-risk molecular-profile BC.

Our study is the first trial to confirm the safety and feasibility of this combination. Our results compare with other ultrafractionation trials using EBRT for tumor bed boost either under simultaneous or sequential treatments [38,39,40,41]. At the present time, there are no data that combine these two ways of administering radiotherapy to patients with BC (IORT-photon high dose plus ultrafractionation WBI).

Limitations of the study include the relatively short number of patients included, which compares to other published series of WBI and photon IORT boosts. A longer follow-up period for the studied patients will be needed to assess the efficacy of the proposed protocol in terms of local control and survival.

## 5. Conclusions

In conclusion, our results confirm for the first time that the combination of ultrafractionated WBI after BCS and photon-IORT is a feasible and safe option in patients with early BC.

## Figures and Tables

**Table 1 cancers-16-01105-t001:** Patient´s clinical characteristics.

Variables	Characteristics	Patients (%)
Age	<45-y	8 (11.1%)
	>45–<55-y	23 (32%)
	>55–<65-y	31 (43%)
	>65	10 (13.9%)
T	pTis	6 (8.3%)
	pT1a	15 (20.9%)
	pT1b	35 (48.6%)
	pT1c	16 (22.2%)
N	pN0	72 (100%)
Histology	In situ ductal carcinoma	6 (8.3%)
	Invasive ductal carcinoma	43 (59.7%)
	Invasive lobular carcinoma	23 (32%)
Grade	1	18 (25%)
	2	45 (62.5%)
	3	9 (12.5%)
Surgical margins	<2 mm	18 (25%)
	>2 mm	48 (66.7%)
	Affected *	6 (8.3%)
Extensive Intraductal component	No	28 (38.9%)
	Yes	44 (61.1%)
Lymphovascular invasion	No	27 (37.5%)
	Focal	36 (50%)
	Extensive	9 (12.5%)
Hormone Receptor	Positive	70 (97.2%)
	Negative **	2 (2.8%)
Menopausal status	pre-menopausal	35 (48.6%)
	post-menopausal	37 (51.4%)
Molecular profile	In situ ductal carcinoma	6 (8.3%)
	Luminal A	50 (69.4%)
	Luminal B (HER2-)	16 (22.2%)
Endocrine therapy	Tamoxifen	20 (27.8%)
	Tamoxifen + Goserelin	16 (22.2%)
	Aromatase inhibitors	34 (47.2%)
	No treatment **	2 (2.8%)

* Affected or focally affected margin. Patients with affected margins underwent surgery to widen the margins (2). ** Hormone receptor-negative in situ ductal carcinoma.

**Table 2 cancers-16-01105-t002:** Acute toxicity *.

Acute Toxicity Assessed with CTCAE v5.0 Grading System				
1st month				
	Grade 0	Grade 1	Grade 2	Grade 3
Pain **	45	18	7	2
Dermatitis (eczema)	52	9	7	4
Pruritus	46	13	9	4
Asthenia	68	3	1	0
3rd month				
Pain **	64	8	0	0
Dermatitis (eczema)	68	4	0	0
Pruritus	71	1	0	0
Hyperpigmentation	60	8	4	---
Skin induration	53	11	8	0
Telangiectasia	67	3	2	---
Asthenia	72	0	0	0

* Most predominant adverse effect is determined. ** Both the breast and the chest wall muscles.

**Table 3 cancers-16-01105-t003:** Late toxicity.

Late Toxicity Assessed with CTCAE v5.0 Grading System				
6th month				
	Grade 0	Grade 1	Grade 2	Grade 3
Pain	69	3	0	0
Fibrosis	59	11	2	0
Skin edema	55	15	2	0
Telangiectasia	66	6	0	---
Ulceration	72	0	0	0
Fat necrosis	65	7	0	0
12th month *				
Pain	66	0	1	0
Fibrosis	62	4	1	0
Skin edema	64	2	1	0
Telangiectasia	65	2	0	---
Ulceration	67	0	0	0
Fat necrosis	62	5	0	0
24th month **				
Pain	50	0	1	0
Fibrosis	47	4	0	0
Skin edema	48	3	0	0
Telangiectasia	50	1	0	---
Ulceration	51	0	0	0
Fat necrosis	48	3	0	0

* 93% (67) of patients. ** 71% (51) of patients.

**Table 4 cancers-16-01105-t004:** Reported characteristics of several trials with Ultrafractionation Whole breast irradiation + Boost.

Autor	Patients	WBI Dose Gy/fx	Dose Boost Gy	Modality Boost	Acute Toxicity
HA15 [38]	95	28.5/5	32.5/34.5	SIB	17.6%-G2–3
YO-HA15 [39]	105	28.5/5	31	SIB	No³G3
Montero [40]	383	26/5	29/30–31	SIB	48%-G1 4%-G2
Machiels [41]	102	26/5	6	Sequential	74%-G1 2.7%-G2
Present trial	72	26/5	20	IORT	No³G3

## Data Availability

The data presented in this study are available on request from the corresponding author.

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
