# Peer review of "Combined Ultrahypofractionated Whole-Breast Irradiation and IORT-Boost: A Safety and Feasibility Analysis"

_cancers, 2024, doi:10.3390/cancers16061105_

Round 1
Reviewer 1 Report
Comments and Suggestions for Authors
This study presents an innovative approach to breast cancer (BC) treatment, focusing on the combination of ultrahypofractionated whole-breast irradiation (WBI) and intraoperative radiotherapy (IORT) boost in patients undergoing breast conserving surgery (BCS). This research is attempting to integrate these modalities, aiming to improve local control while minimizing treatment time and potential toxicity.
The methodology involves 72 patients diagnosed with low-risk early BC, eligible for BCS, treated with a regimen that combines a photon-IORT boost at the time of surgery followed by ultrahypofractionated WBI.
The study outlines the treatment protocol, including patient selection criteria, treatment delivery specifics, and follow-up measures. The primary endpoints are safety and feasibility, with secondary endpoints addressing cosmetic outcomes, local relapse rate, and survival.
Results indicate that the treatment protocol was feasible, with the majority of patients receiving the planned treatments within the stipulated timeframe. Acute and late toxicities were within acceptable limits, with no severe acute or late toxicities reported. Cosmetic outcomes were generally favorable, and no disease recurrence was observed within the follow-up period.
I appreciate the study's novelty; however, some modifications are recommended to enhance the manuscript's clarity and comprehensiveness:
-
- The simple summary is missing;
-
- The abstract should be divided into: background, methods, results, and conclusions. Additionally, limitations do not belong into the abstract;
-
- In the introduction section (lines 37-38) it is certainly true that "Unfortunately, most local recurrences of BC have been observed to occur at the orig inal primary site [10-12]"; in fact, it is essential to incorporate additional literature to contextualize the significance of local recurrences in BC, specifically recommending the inclusion of PMID: 33431329, which investigates ipsilateral breast cancer recurrences;
-
- The limitations section, currently brief and placed at the end, should be expanded to thoroughly discuss the study's constraints, focusing on the small sample size and the relatively short follow-up period. This section should precede the conclusions.
The manuscript would benefit from a moderate language revision to correct typos, misspellings, and grammatical errors.
Author Response
1.-The simple summary is missing;
Response: we add a simple summary. Now reads:
The aim of the present study is to assess for the first time the feasibility and safety of combining photon IORT with ultrahypofractionated WBI after breast conserving surgery (BCS). From July 2020 to December 2022, seventy-two patients diagnosed of low risk early BC were included in this prospective study. IORT was administered to a dose of 20 Gy to the surface´s applicator and WBI was to be administered 3-5 weeks after surgery to a total dose of 26 Gy in 5 consecutive days. All patient completed the proposed treatment schedule and no severe acute or late grade 3 toxicity was observed 3 and 12 months after WBI respectively. Our results confirm for the first time, that the combination of ultrafractionation WBI after BCS and photon-IORT, is a feasible and safe option in patients with early BC.
2.- The abstract should be divided into: background, methods, results, and conclusions. Additionally, limitations do not belong into the abstract;
Response: The abstract has been modified as suggested. Now reads:
Abstract:
Background: The current standard of local treatment for patients with localized breast cancer (BC) includes whole breast irradiation (WBI) after breast conserving surgery (BCS). Ultrahypofraction-ated WBI schemes (1 week treatment) were shown not to be inferior to the standard WBI. Tumor bed boost using photon intraoperative radiotherapy (IORT) has been shown to be safe and feasible in combination to standard WBI. The aim of the present study is to assess for the first time the fea-sibility and safety of combining photon IORT with ultrahypofractionated WBI.
Methods :Patients diagnosed of low-risk early BC, candidates for BCS were included in this pro-spective study. IORT was administered to a dose of 20 Gy to the surface´s applicator and WBI was to be administered 3-5 weeks after surgery to a total dose of 26 Gy in 5 consecutive days.
Results: From July 2020 to December 2022, seventy-two patients diagnosed of low risk early BC and treated in our institution, were included in this prospective study. All patient completed the proposed treatment schedule and no severe acute or late grade 3 toxicity was observed 3 and 12 months after WBI respectively.
Conclusions: our results confirm for the first time, that the combination of ultrafractionation WBI after BCS and photon-IORT, is a feasible and safe option in patients with early BC.
3.- In the introduction section (lines 37-38) it is certainly true that "Unfortunately, most local recurrences of BC have been observed to occur at the original primary site [10-12]"; in fact, it is essential to incorporate additional literature to contextualize the significance of local recurrences in BC, specifically recommending the inclusion of PMID: 33431329, which investigates ipsilateral breast cancer recurrences;
Response: The suggested reference has been included
Sagona A, Gentile D, Anghelone CAP, Barbieri E, Marrazzo E, Antunovic L, Franceschini D, Tinterri C.Ipsilateral Breast Cancer Recurrence: Characteristics, Treatment, and Long-Term Oncologic Results at a High-Volume Center. Clin Breast Cancer. 2021 Aug;21(4):329-336. doi: 10.1016/j.clbc.2020.12.006. Epub 2020 Dec 17
4.- The limitations section, currently brief and placed at the end, should be expanded to thoroughly discuss the study's constraints, focusing on the small sample size and the relatively short follow-up period. This section should precede the conclusions
Response: we followed the reviewer´s suggestions. Now reads:
Limitations of the study include the relatively short number of patients included, that compares to other published series of WBI and photon IORT boost. A longer follow-up period of the studied patients will be needed to assess efficacy in terms of local control and survival of the proposed protocol.
Reviewer 2 Report
Comments and Suggestions for Authors
I read the interesting manuscript from Javier Burgos-Burgos et al.
Please find below my comments:
- Introduction: please better explain the details of the different radiotherapy treatments cited in the manuscript (conventionally fractionated radiotherapy VS those from the moderate hypofractionation- and the ultrafractionation Fast Forward- trials). Please also shortly introduce the concept of the radiotherapy- and boost- combination.
- Please move the “patients characteristics” (and Table 1) paragraph in the materials and methods section. These do not belong to the results part. Also, please divide the “treatment” paragraph (Materials and Methods) into two parts: the treatment description and the evaluation.
- In the Discussion part there is a nice description of the different approaches to boost treatment when using ultrafractionation schedules. Could you include and comment on the results of the other trials combining radiation therapy and IORT? This is necessary to understand the results of your trial.
Author Response
Reviewer 2
1.-Introduction: please better explain the details of the different radiotherapy treatments cited in the manuscript (conventionally fractionated radiotherapy VS those from the moderate hypofractionation- and the ultrafractionation Fast Forward- trials). Please also shortly introduce the concept of the radiotherapy- and boost- combination.
Response: we followed the reviewer´s suggestions. Now reads:
This treatment approach has been shown to be equivalent to mastectomy in terms of local control and survival [2-4]. Although, conventionally fractionated radiotherapy (50Gy 2Gy per fraction) administered in 25 fractions during 5 weeks radiotherapy was historically used, clinical evidence from moderate hypofractionation trials [5-7] with shorter treatment times and higher doses per session, based on the low alpha/beta (3Gy) of BC [8] gave favorable results. These moderated hypofractionation schemes include 40.05 Gy at 2,67 Gy per fraction, during a total of 15 fractions administered in 3 weeks. The increased local control and reduced toxicity rates observed in moderate hypofractionation trials [5-7], encouraged researchers to try to further increase the doses per session, further shortening the total treatment time. In May 2020, the results of the ultrafractionation Fast Forward trial [9] were released, showing that the 26Gy/5,2Gy per fraction, during 5 fractions administered in only 1week, scheme was not inferior to the standard 40.05Gy/15fx/3w for local tumor control and is equally safe in terms of effects on normal tissue.
Unfortunately, most local recurrences of BC have been observed to occur at the original primary site [10-12]. An additional dose of radiation to the tumor bed (boost radiation dose) is accompanied by an increase in local control for patients at the expense of greater toxicity depending on the technique used [13]. According to this evidence, WBI+boost dose would be considered the standard treatment for breast cancer patients after BCS.[13]
2.- Please move the “patients characteristics” (and Table 1) paragraph in the materials and methods section. These do not belong to the results part. Also, please divide the “treatment” paragraph (Materials and Methods) into two parts: the treatment description and the evaluation.
Response: We followed the reviewer´s suggestion. Now the Materials and Methods section now reads
Materials and Methods
Patients diagnosed of low-risk early BC, candidates for BCS were included in this prospective study. Patients were diagnosed and treated at the Canarian Comprehensive Cancer Center, San Roque University Hospital in Las Palmas de Gran Canaria - Spain. In all cases, the approval of the Ethics Committee and written informed consent were obtained. Patient´s inclusion criteria were: age 18 years or older, histologically and radiologically proven early BC, tumor size less than 2cm, Luminal molecular profile, and must be discussed in Multidisciplinary Tumor Boards (MTB) of our institution. Exclusion criteria were: age less than 18 years, histological types other than epithelial breast cancer, axillary positive nodes, locally advanced or metastatic disease or ECOG (Eastern Cooperative Oncology Group) >3. Cancer staging was performed according to the 8th edition of the TNM classification system [17].
WBI was prescribed to those patients ≤ 45 years of age and those who presented in their pathology report infiltrative lobular histology, pure intraductal carcinoma in situ, extensive intraductal component, lymphovascular invasion and close or affected surgical margins. From July 2020 to December 2022, seventy-two patients were included in the study. The clinical characteristics of the patients are shown in Table 1. The median age was 56 years (range 36-72 years). The size of the tumor was less than 2 cm in all cases. Thirty-two patients presented premenopausal and forty patients postmenopausal status receiving the corresponding endocrine therapy according to the international guidelines [18,19]. No patient was considered for chemotherapy.
Table 1. Patients clinical characteristics.
|
Variables |
Characteristics |
Patients (%) |
|
Age |
< 45-y >45 - <55-y >55 - <65-y >65 |
8 (11.1%) 23 (32%) 31 (43%) 10 (13.9%) |
|
T |
pTis pT1a pT1b pT1c |
6 (8.3%) 15 (20.9%) 35 (48.6%) 16 (22.2%) |
|
N |
pN0 |
72 (100%) |
|
Histology |
In situ ductal carcinoma Invasive ductal carcinoma Invasive lobular carcinoma |
6 (8.3%) 43 (59.7%) 23 (32%) |
|
Grade |
1 2 3 |
18 (25%) 45 (62.5%) 9 (12.5%) |
|
Surgical margins |
< 2 mm >2mm Affected* |
18 (25%) 48 (66.7%) 6 (8.3%) |
|
Extensive Intraductal component |
No Yes |
28 (38.9%) 44 (61.1%) |
|
Lymphovascular invasion |
No Focal Extensive |
27 (37.5%) 36 (50%) 9 (12.5%) |
|
Hormone Receptor |
Positive Negative** |
70 (97.2%) 2 (2.8%) |
|
Menopausal status |
pre-menopausal post-menopausal |
35 (48.6%) 37 (51.4%) |
|
Molecular profile |
In situ ductal carcinoma Luminal A Luminal B (HER2-) |
6 (8.3%) 50 (69.4%) 16 (22.2%) |
|
Endocrine therapy |
Tamoxifen Tamoxifen+Goserelin Aromatase inhibitors No treatment** |
20 (27.8%) 16 (22.2%) 34 (47.2%) 2 (2.8%) |
* Affected or focally affected margin. Patients with affected margins underwent surgery to widen the margins (2). ** Hormone receptor-negative in situ ductal carcinoma.
Treatment description
All patients were referred to BCS and immediate IORT at the time of lumpectomy. Hormonal therapy was prescribed according to international guidelines [18,19]. The IORT treatment in our institution has been previously described [15,16]. In short, a radiation dose of 20Gy was prescribed to the surface of the applicator [15,16] of a portable low-energy X-ray accelerator (50kv), Intrabeam® (Carl-Zeiss Ober Kochen Germany). Postoperative whole breast irradiation (WBI) was administered to patients showing the TARGIT A adverse factors present in the pathology report [20]. The WBI dose to be administered was that of the ultrafractionation scheme (26Gy/5.2Gy/5fx) of the Fast Forward study [9] and it was planned to be administered 3 to 5 weeks after BCS.
WBI treatment after IORT in our institution has been previously described [15,16]. All patients were treated by the VMAT-Rapid Arc technique. CTVs and OARs contouring were based on the recommendations of RTOG [21]. The same OARs constraints of the reference FAST-FORWARD study were used [9]. The plans were optimized for Rapid Arc delivery and the PRO algorithm [22] was used to modulate the shape of the multi-leaf collimator and the intensity of the beam during gantry rotation. Dose calculations used the anisotropic analytical algorithm (AAA).
The delivery was made by a VARIAN Linac, using 6MV photons, (Palo Alta, California). Daily image-guided radiation therapy (IGRT) using an Exac-Trac System (Brain Lab, Munich, Germany) was performed on all patients.
In addition, all patients were offered to receive a comprehensive program of nutrition counseling, physical rehabilitation, psycho-oncology support and dermo-aesthetics therapy with the aim of improving functionality and patient´s quality of life.
Evaluation
The defined primary end points of the study were safety and feasibility of the proposed treatment protocol. Safety was evaluated by the rate of grade 3 acute and late toxicity observed in the treated patients, according to the common toxicity criteria adverse effects (CTCAE 5.0) [23]. Patients were evaluated for toxicity one, three and six months after the end of radiotherapy. Patients were followed prospectively under the Spanish RD1566/1998 regulation for Radiation Therapy Quality Assurance.
Follow-up visits were subsequently scheduled every 3 months during the first two years, and every six months, thereafter. Breast Ultrasound and mammograms were performed six months after BCS and yearly thereafter, otherwise recommended. Follow-up was jointly performed by Surgery, Medical and Radiation Oncology staff doctors of the Canarian Comprehensive Cancer Center.
The treatment feasibility as evaluated through the number of patients that completed the scheduled treatment and the number of patients that suffered interruptions of the treatment.
The defined secondary endpoints included, cosmetic results evaluated by the Harvard scale [24], local relapse rate, and cause specific/overall survival.
3.- In the Discussion part there is a nice description of the different approaches to boost treatment when using ultrafractionation schedules. Could you include and comment on the results of the other trials combining radiation therapy and IORT? This is necessary to understand the results of your trial.
Response: we followed the reviewer´s comments. Now reads:
Conventionally fractionated WBI (50 Gy in 5 weeks) combined with IORT photon boost was shown to be an standard alternative to other boost techniques [29]. Moderate hypofractionation WBI (40.05 Gy in 3 weeks) combined with IORT boost has been con-firmed to be safe and feasible using both electrons [30-33] and photons [15,16,34,35]. Our pioneering experience has shown that the combination of immediate IORT plus moderate HWBI is accompanied by favorable results in relation to chronic toxicity and aesthetic results both in patients with early and locally advanced disease [15,16].
[29]Blank, E., Kraus-Tiefenbacher, U., Welzel, G. et al. Single-Center Long-Term Follow-Up After Intraoperative Radiotherapy as a Boost During Breast-Conserving Surgery Using Low-Kilovoltage X-Rays. Ann Surg Oncol 17 (Suppl 3), 352–358 (2010). https://doi.org/10.1245/s10434-010-1265-z